# Dual-Hit Model of Parkinson’s Disease: Impact of Dysbiosis on 6-Hydroxydopamine-Insulted Mice—Neuroprotective and Anti-Inflammatory Effects of Butyrate

**DOI:** 10.3390/ijms23126367

**Published:** 2022-06-07

**Authors:** Carmen Avagliano, Lorena Coretti, Adriano Lama, Claudio Pirozzi, Carmen De Caro, Davide De Biase, Luigia Turco, Maria Pina Mollica, Orlando Paciello, Antonio Calignano, Rosaria Meli, Francesca Lembo, Giuseppina Mattace Raso

**Affiliations:** 1Department of Pharmacy, School of Medicine, University of Naples Federico II, Via Domenico Montesano, 80131 Naples, Italy; carmen.avagliano@unina.it (C.A.); lorena.coretti@unina.it (L.C.); claudio.pirozzi@unina.it (C.P.); luigia.turco@unicampania.it (L.T.); calignan@unina.it (A.C.); meli@unina.it (R.M.); frlembo@unina.it (F.L.); 2Task Force on Microbiome Studies, University of Naples Federico II, Via Domenico Montesano, 80131 Naples, Italy; mpmollic@unina.it; 3Department of Science of Health, School of Medicine, University Magna Graecia of Catanzaro, Viale Europa, 88100 Catanzaro, Italy; decaro@unicz.it; 4Department of Pharmacy, University of Salerno, 84084 Fisciano, Italy; davide.debiase@hotmail.com; 5Department of Precision Medicine, University of Campania Luigi Vanvitelli, 80138 Naples, Italy; 6Department of Biology, University of Naples Federico II, Complesso Universitario di Monte Sant’Angelo, Cupa Nuova Cinthia 21 Edificio, 80126 Naples, Italy; 7Department of Veterinary Medicine and Animal Production, University of Naples Federico II, Via Delpino, 80137 Naples, Italy; paciello@unina.it

**Keywords:** neurodegenerative disorders, gut microbiota, antibiotic-induced intestinal injury, short-chain fatty acids, neuroinflammation

## Abstract

Recent evidence highlights Parkinson’s disease (PD) initiation in the gut as the prodromal phase of neurodegeneration. Gut impairment due to microbial dysbiosis could affect PD pathogenesis and progression. Here, we propose a two-hit model of PD through ceftriaxone (CFX)-induced dysbiosis and gut inflammation before the 6-hydroxydopamine (6-OHDA) intrastriatal injection to mimic dysfunctional gut-associated mechanisms preceding PD onset. Therefore, we showed that dysbiosis and gut damage amplified PD progression, worsening motor deficits induced by 6-OHDA up to 14 days post intrastriatal injection. This effect was accompanied by a significant increase in neuronal dopaminergic loss (reduced tyrosine hydroxylase expression and increased Bcl-2/Bax ratio). Notably, CFX pretreatment also enhanced systemic and colon inflammation of dual-hit subjected mice. The exacerbated inflammatory response ran in tandem with a worsening of colonic architecture and gut microbiota perturbation. Finally, we demonstrated the beneficial effect of post-biotic sodium butyrate in limiting at once motor deficits, neuroinflammation, and colon damage and re-shaping microbiota composition in this novel dual-hit model of PD. Taken together, the bidirectional communication of the microbiota–gut–brain axis and the recapitulation of PD prodromal/pathogenic features make this new paradigm a useful tool for testing or repurposing new multi-target compounds in the treatment of PD.

## 1. Introduction

The classic view of Parkinson’s disease (PD) as a merely motor disorder, involving exclusively CNS structures, has progressed in the last two decades based on non-motor signs characterizing a long prodromal period, associated with extra-nigral sites [1]. These preceding ailments include olfactory dysfunction and gastrointestinal symptoms (i.e., constipation, delayed gastric emptying, and oesophageal dysfunction) [2], indicating a complex pathogenetic mechanism that over time converges toward a full-blown disease. Among extra-nigral areas, different evidence pinpointed the involvement of a network of different structures through which PD progresses, from the olfactory bulb and enteric nerve cell plexus, the major gateways for the environmental factors, up to the substantia nigra and other brain areas via the vagus nerve [3,4]. This hypothesis was originally postulated by Braak et al. [5] as a bottom-up scenario, where the early accumulation of misfolded α-synuclein-containing inclusions, named Lewy bodies, in the gut spreads in a centripetal fashion through trans-synaptic cell-to-cell transmission to cauda-rostral areas of the brain, leading to progressive degeneration of dopaminergic neurons in the substantia nigra pars compacta.

The gut-brain neural connections have evolved to comprise further integrative messengers including gut microbiota and their products. In the last few years, several studies highlighted the pivotal role of the gut microbiota as a key regulator of immunological and neuroendocrine mechanisms of the gut–brain axis in PD [6,7]. In this context, research efforts had been focused on the understanding of the pathogenic mechanisms associated with neurodegeneration and specifically with a dysfunctional gut–brain axis, underlying PD onset, development, and progression. Therefore, many PD experimental models have been extensively fine-tuned to recapitulate PD etiopathology and decipher the intricate connections among microbiota, gut, and brain [8,9,10], as well as discover new therapeutic tools to be efficaciously translated into clinical applications. Notably, to date, the most effective symptomatic treatment for Parkinson’s disease (PD) remains levodopa/carbidopa, although new emerging therapeutics and adjunctive nutraceuticals have been proposed [11,12]. The lack of full translation of experimental findings in PD patients has highlighted key drawbacks of simplified animal models in recapitulating the complex and degenerative human disease, opening an interesting scientific debate [13].

Recent advances strongly suggested the correlation between neuroinflammation, neurodegeneration, and gut microbiota alteration [14], sustained by host genomics, lifestyle, age, sex, comorbidity, and polypharmacy [15]. Hence, prolonged dysbiosis leads to gut immune dysregulation, intestinal barrier disruption, and local and/or systemic inflammation, with the potential to perturb brain homeostasis [16]. To corroborate the important role of gut microbiota in PD pathology, very recent studies have demonstrated that modulation of gut microbiota could worsen or ameliorate motor symptoms. Recently, it has been reported that in 1-methyl-4-phenyl-1,2,3,6-tetrahydropyridine (MPTP)-induced PD mice, fecal microbiota transplantation successfully reduced motor deficits and dopaminergic neuronal damage [10]; moreover, the probiotic strain Lactobacillus plantarum PS128 alleviated the neurodegenerative progression and neurobehavioral impairment of PD [17]. Recent studies in animal models have demonstrated that the modulation of gut microbiota by probiotic or post-biotic treatments could ameliorate motor symptoms, alleviating the neurodegeneration and counteracting the neurobehavioral impairment of PD [17,18]. Gut microbiota alteration in PD patients, as well as animal models, has been associated with a reduction in those bacteria producing short-chain fatty acids (SCFAs) [17,19,20,21,22,23,24]. Therefore, the shift in gut microbiota associated with PD leads to a change in microbial metabolite production, underlying gastrointestinal and motor dysmotility in PD. It has been proven how the complexity and diversity of the microorganisms that inhabit the gut, as well as their produced metabolites, can regulate PD onset and progression [25], reinforcing the evidence that PD is a neurodegenerative disorder involving both gut and brain. Among SCFAs, butyrate exerts multiple intestinal effects, such as the reduction of inflammation and oxidative status, strengthening of the defense barrier, as well as the modulation of visceral sensitivity and bowel motility. Different mechanisms of action have been proposed, including epigenetic modifications through inhibition of histone deacetylases [26], inhibition of NF-kB signaling [27], and direct binding and activation of the free fatty acid receptor (FFAR)-2 and 3 (GPR41 and GPR43, respectively) [28].

In the present study, we have designed a two-hit model of PD to investigate the mechanisms related to gut dysbiosis and inflammation in PD progression. Specifically, we examined how ceftriaxone-induced dysbiosis and gut alterations interacted and influenced PD onset and progression in 6-hydroxydopamine (6-OHDA)-insulted mice. We also investigated the effect of a post-biotic treatment with sodium butyrate (BuNa) on behavioral alterations and biochemical modifications related to neuroinflammation and neurodegeneration, as well as on gut damage and microbiota composition.

## 2. Results

### 2.1. Motor Coordination and Apomorphine-Induced Rotational Behavior in CFX Dysbiotic PD Mice. BuNa Effect on Behavioral Traits

The rotarod was used to evaluate the motor coordination of mice on a rolling rod after 3, 7, and 14 days post-surgery (Figure 1B-D). The time spent on the rotating rod was significantly decreased 3 or 7 days after the 6-OHDA challenge when compared with the Sham group (Figure 1B,C, respectively), while CFX pretreatment did not affect motor impairment of 6-OHDA mice. Conversely, at 14 days, the motor deficit of 6-OHDA mice was recovered (Figure 1D), as already reported [29]. Notably, motor coordination was not restored when 6-OHDA-injected mice were pretreated with CFX, compared to Sham or 6-OHDA-injected mice, indicating persistence of motor impairment upon CFX challenge (Figure 1D). BuNa treatment over time improved motor coordination at day 7 both in 6-OHDA and dual-insulted mice (Figure 1C,D).

Behavioral quantification of DA depletion was measured by apomorphine-induced rotations at 3, 7, and 14 days post-surgery (Figure 1E–G). Total rotations induced by the apomorphine in the 6-OHDA group were significantly increased compared to the Sham group and were further augmented by antibiotic treatment at all time points, indicating a worsening effect of CFX on striatal dopaminergic loss. BuNa administration ameliorated 6-OHDA-induced neurobehavioral deficit as early as within 7 days of treatment (Figure 1F), and its effects on apomorphine-induced rotational behavior were even more evident in CFX-treated PD mice at 7 and 14 days (Figure 1F–G). Both behavioral tests performed on CFX-treated animals showed similar results to those of Sham-operated mice, indicating no effect of CFX alone on motor function.

### 2.2. 6-OHDA-Induced Striatal Toxicity Is Worsened by CFX Pretreatment. Reversal Effect of BuNa

To confirm striatal neurodegeneration, the expression of tyrosine hydroxylase (TH), the rate-limiting enzyme in DA synthesis, was assessed by Western blot analysis. The reduction of TH expression in the lesioned striatum of 6-OHDA mice was even more profound when the mice were pre-treated with CFX, indicating higher neurodegeneration in dysbiotic mice (Figure 2A). BuNa treatment of 6-OHDA mice prevented TH reduction, and to a lesser extent did the same in 6-OHDA+CFX mice. Western blot analysis of striatal lysates also showed an increase in nNOS and iNOS expression that was similar in both 6-OHDA-insulted mice (6-OHDA and 6-OHDA+CFX groups) (Figure 2B,C). The increased striatal expression of COX-2 in 6-OHDA mice, instead, was significantly higher in PD mice pretreated with CFX (Figure 2D). BuNa treatment reduced the expression of all of the inflammatory enzymes in both the 6-OHDA and 6-OHDA+CFX groups (Figure 2B–D). No changes in nNOS, iNOS and COX-2 expression were shown in striata of the CFX group. Moreover, to investigate the neuroprotective effect by which BuNa may limit the development of neurotoxicity by 6-OHDA, we evaluated the ratio between anti-apoptotic Bcl-2 and pro-apoptotic Bax expression, demonstrating a reduction of apoptosis in BuNa-treated animals (Figure 2E). Indeed, the immunoreactivity for Bcl-2 was increased, whereas Bax expression was significantly decreased in striata from BuNa-treated groups, reverting the lowered ratio found in untreated 6-OHDA- or 6-OHDA+CFX-challenged mice.

### 2.3. Alteration of Systemic Pro-Inflammatory Markers. Protective Effect of BuNa

Neuroinflammation has been associated with the progression and severity of PD. In parallel, inflammatory cytokines were elevated at the systemic level (Figure 3). 6-OHDA mice showed a significant increase in serum TNF-α, IL-1β, and IFN-γ compared to Sham mice (Figure 3A–C), and levels of those cytokines, except for IL-1β, were even more increased in CFX-pretreated PD mice. Interestingly, an increased level of cytokines was also shown in the CFX group, indicating a clear contribution of colonic damage to systemic inflammation. Moreover, only CFX-treated mice, with or without the 6-OHDA challenge, showed an increase in endotoxemia (Figure 3D), suggesting the transfer of macromolecules, such as microbial- or pathogen-associated molecular patterns (i.e., lipopolysaccharide, LPS) into the systemic circulation. BuNa treatment limited systemic inflammation and endotoxemia in 6-OHDA+CFX mice.

### 2.4. Colon Injury in Dysbiotic PD Mice and BuNa Effect on Gut Homeostasis

Histology was performed on colon samples of dysbiotic PD mice in order to assess colon injury. The sections were graded by two blinded investigators (O.P. and D.D.B.) with a range from 0 to 4 for the amount of tissue damage, with a range from 0 to 3 for the extent of tissue damage, with a range from 0 to 3 for the degree of inflammation, and with a range from 0 to 4 for the amount of crypt damage or regeneration. The final histological score was established by summing the scores for each parameter.

From histological analysis (Figure 4A), the distal colons of the Sham control group showed no significant pathological alteration, and the mean value of the histological score was 1. 6-OHDA mice showed significant structural abnormalities, such as moderate to marked villi attenuation, occasional syncitia of villous enterocytes, crypt degeneration, and inflammatory cell infiltration appearing mostly in lymphocytes and plasma cells. The mean value of the histological score for the 6-OHDA group was 4.6. Morphological abnormalities were worsened in CFX pre-treated mice with or without the 6-OHDA challenge. The mean values of the histological score were 7.6 for the 6-OHDA+CFX group and 5 for CFX group. Histological assessment of colon sections upon treatment with BuNa of 6-OHDA or 6-OHDA+CFX mice revealed an improved morphological outcome, ameliorating histological changes and limiting intestinal inflammation and tissue damage. The mean values of the histological score for the 6-OHDA+BuNa group and 6-OHDA+CFX+BuNa group were respectively 2.3 and 4.6.

Consistently, we evaluated the mRNA expression of IL-1β, COX-2, and IL-10 (Figure 4B–D) in the colon of all groups. Interestingly, 6-OHDA injection alone triggered colonic inflammation, while pretreatment with CFX further increased *Il1b* transcription. BuNa treatment limits these detrimental alterations induced by 6-OHDA and/or CFX, increasing *Il10* and reducing *Il1b* and *Ptgs2* mRNAs.

### 2.5. Antibiotic-Induced Dysbiosis in 6-OHDA-Lesioned Mice and BuNa Effect on Microbiota Composition

Using targeted sequencing of the 16S rRNA gene, we profiled the structure of the fecal microbiota of 6-OHDA mice pretreated with CFX and compared it to those of the Sham, 6-OHDA-challenged, and CFX groups. An average number of 8288.45 reads/sample (minimum frequency of 2344 reads) were acquired from fecal samples by using Illumina sequencing after sequence denoising, trimming, and chimera picking. These reads were clustered into 3063 amplicon sequence variants (ASVs); the rarefaction curve, showing that all samples reached their plateaus, and a mean Good’s Coverage over 99% suggested the adequacy of the sequencing.

The gut microbiota analysis of 6-OHDA mice and Sham mice at 14 days after surgery showed a slight difference in gut microbial composition between the two groups (ANOSIM on unweighted UniFrac distances R = 0.189 *p* = 0.042; Appendix A). LEfSe analysis at the genus level, taxonomically implemented using BLAST NCBI Database, revealed that the observed difference in microbial assortment between the two groups was mainly based on a higher content of *Prevotellamassilia timonensis* and an Unclassified species (U.s.) of Eubacteriales and a lower abundance of a U.s. of *Ligilactobacillus* in 6-OHDA mice compared to Sham control mice (Appendix A).

Pretreatment with CFX of 6-OHDA mice significantly decreased the number of observed ASVs and their intra-group distribution and changed microbial composition at the genus level as of 3 days of antibiotic exposure compared to untreated 6-OHDA and Sham mice (Appendix A). Notably, analysis at 14 days from antibiotic treatment highlighted enduring reduced richness in the 6-OHDA+CFX group not only compared to the 6-OHDA and Sham groups but also to the CFX group (Appendix A). These data were corroborated by core microbiota analysis at the genus level among all groups at 14 days from antibiotic treatment. Indeed, 6-OHDA+CFX mice shared the lowest number of bacterial genera with Sham compared to the other groups; among the 26 missing genera, seven genera were specifically absent in the 6-OHDA+CFX group while being shared by the Sham, CFX, and 6-OHDA groups (Figure 5A). LEfSe analysis showed that 6-OHDA+CFX mice harbored a microbiota depleted in the relative abundance of *Odoribacter*, *Bacteroides*, *Prevotella*, *Defluviitalea* (*saccharophila*), *Papillibacter* (*cinnamivorans*), and members of Lachnospiraceae and S24-7 families compared to the Sham and 6-OHDA groups. Particularly, the genera *Bacteroides* and *Defluviitalea* were also significantly reduced in 6-OHDA+CFX mice compared to CFX mice. The single increased bacterial genus in the 6-OHDA+CFX group was *Ruminococcus* (*lactaris*) whose relative abundance raised from 1% in Sham, 1.01% in 6-OHDA, and 0.59% in CFX mice up to 3.27% in 6-OHDA+CFX mice (Figure 5B–D). The representative sequence of each ASV belonging to key genera was also aligned to the NCBI 16S BLAST database to increase taxonomic resolution (see Methods and Supplementary Dataset 1). The resulting species were studied in a correlation network to identify patterns of co-occurring bacteria in the complexity of microbial interactions among all the groups (Figure 6). The network contained 46 nodes (microbial species) connected by 31 significant edges, and the clustering patterns observed indicated the groups of co-occurring taxa within microbiota communities. *Papillibacter cinnamivorans* and *Anaerorhabdus furcosa* were the species with the highest numbers of edges, tending to co-occur with 16 and 15 bacterial species, respectively, and both negatively correlated with *Ruminococcus lactaris. R. lactaris* also showed mutual exclusion with *Bifidobacterium breve, Alistipes finegoldii*, and U.s of S24-7. Moreover, the betweenness centrality feature was selected to measure the centrality of each node in the network. Consistently with the afore-described co-occurring taxa interaction, the highest betweenness centrality scores were observed for *A. furcosa* and *P. cinnamivorans* and secondly for *Defluviitalea saccarophila*, indicating these species as the keystone bacteria driving the positively connected species. Collectively, the data showed that the reduction of *A. furcosa, P. cinnamivorans*, and *D. saccarophila*, together with the increase of *R. lactaris*, drive the microbiota dysbiosis induced by antibiotic treatment of 6-OHDA mice.

Definition of gut microbiota structure was obtained at 14 days of treatment with BuNa in CFX pre-treated 6-OHDA mice. Remodeling of 6-OHDA+CFX bacterial communities upon the postbiotic treatment was observed in control and 6-OHDA+CFX mice. Specifically, 21 shared bacterial genera were now observed as a result of core microbiota analysis between the 6-OHDA+CFX+BuNa and Sham groups as being several bacterial genera distinctive of 6-OHDA+CFX+BuNa (*Lactobacillus hamsteri*, *Odoribacter*, *P. cinnamivorans*, *D. saccharophila*, *Dorea longicatena*, *Clostridium saccharogumia* and U.g. of Firmicutes; see Methods for species-level resolution and Supplementary Dataset 1) and absent in 6-OHDA+CFX mice (Figure 7A). LEfSe analysis also showed a reduction of *Ruminococcus (lactaris)*, *Oscillospira (guilliermondii)*, and an unclassified genus of the Micrococcaceae family upon BuNa treatment compared to 6-OHDA+CFX mice (Figure 7B). However, a new microbial balance was established in 6-OHDA+CFX+BuNa mice compared to Sham mice; enrichment of *Akkermansia muciniphyla* and reduction of *Paraeggerthella (hongkongensis)*, *Butyricicoccus (pullicaecorum)*, *Oscillospira (guilliermondii)*, *Desulfovibrio (alaskensis)*, members of Micrococcaceae, Bacteroidales, YS2, Clostridiales, and Rhodobacteraceae marked 6-OHDA+CFX+BuNa mice, defining a new distinctive signature of their microbiota (Figure 7B).

## 3. Discussion

The present study demonstrates that CFX-induced gut dysbiosis in mice bridges the peripheral intestinal damage to neuroinflammation in mediating progressive neurodegeneration in PD. The two-hit (dysbiosis and striatal neurotoxin challenge) animal model proposed in this study aims to mimic PD multifactorial etiology and reproduces several key features of PD, underlying the “synergistic” neurotoxicity deriving from striatal 6-OHDA injection in CFX-treated dysbiotic mice. Here, we showed that CFX pretreatment before striatal challenge induced an exacerbated and progressive motor deficit and increased striatal degeneration of dopaminergic neurons, indicating a peripheral detrimental contribution to neurodegeneration.

Indeed, the 6-OHDA injection into the striatum is a model of PD characterized by a partial and progressive nigrostriatal neurodegeneration; however, the lack of prodromal features, independent from striatal dopamine deficit, partially limits the construct and/or predictive validity of this PD model. Recently, we characterized an antibiotic-induced intestinal injury, as a dysbacteriosis model in mice, by oral administration of CFX at a high dose [30]. The antibiotic treatment induced a time-dependent (5 and 15 days) increase in systemic inflammation (TNF-α, IL-1β, and IFN-γ) and derangement of intestinal homeostasis due to the loss of colonic architecture and the alteration of gut microbiota composition and diversity [30]. Here, we modified the “classical” 6-OHDA-induced PD model with 5-day CFX pretreatment as a preceding hit to mimic a peripheral prodromal contribution deriving from gut microbiota dysbiosis and intestinal impairment. Therefore, this dual-hit model of PD recapitulates both peripheral and central pathogenic mechanisms and could represent a novel paradigm for testing multi-target therapeutics, acting on the separate but converging pathways leading to neurodegeneration.

Our results indicate that the double insult induced a worsening of the motor deficit compared to the single 6-OHDA challenge. CFX pretreatment resulted in a more prolonged reduction in motor coordination, which lasted 14 days after intrastriatal injection, compared to 6-OHDA mice that completely recovered motor function. In parallel, CFX-pretreated 6-OHDA mice showed at all time points a significant increase in the rotational behavior induced by apomorphine administration, indicating marked dopaminergic neurodegeneration. Consistently, 6-OHDA+CFX mice showed a significant decrease in TH expression in the striatum compared to the 6-OHDA group. The neurodegenerative process was also paralleled by increased striatal neuroinflammation and apoptosis. Notably, CFX pretreatment of 6-OHDA mice synergistically drove a cascade of local-to-systemic inflammation, as shown by the increased cytokine serum levels, and endotoxemia. All of these findings in this two-hit model support the crucial role of interactions among microbiota, the gut, and the brain in PD. Gut microbiota perturbation and intestinal damage by CFX accelerated 6-OHDA-induced striatal neurodegeneration, also sustained by systemic low-grade inflammation.

A recent article by McQuade et al. [31] compared the level of enteric neuropathy and degree of gastrointestinal dysfunction across several commonly used mouse models of PD, indicating that, as well as all those considered, the 6-OHDA model shows some degree of enteric neuropathy without overt gastrointestinal dysfunction. Moreover, 6-OHDA-treated rats were shown to exhibit a significant reduction in daily fecal pellet output [32], longitudinal muscle contraction and intraluminal pressure in the distal colon [33], and gastric emptying and intestinal transit [34]. Previous findings had demonstrated that the peripheral modifications secondary to 6-OHDA injection were likely causing trans-synaptic effects in pathways from the brain sending signals down the spinal cord to the colorectum [35]. In this model the gastrointestinal effects demonstrated a top-down scenario, indicating that nigrostriatal denervation induced significant changes in the gastrointestinal tract functions [36]. Here, mice subjected only to 6-OHDA hit showed villi attenuation, occasional syncytia of villous enterocytes, crypt degeneration, and inflammatory cell infiltration, all structural abnormalities worsened in dual-insulted mice (6-OHDA+CFX group). Moreover, 6-OHDA mice, as well as CFX-treated ones, showed a detrimental unbalance between pro- and anti-inflammatory mediators at the colonic level, which was even more profound in dual-hit mice.

One of the central aims of the proposed two-hit model was to follow the dynamics of gut microbiota upon treatments to infer their possible role in modulating the bottom-up pathways. In this study, local inflammation resulting in systemic translocation of pro-inflammatory molecules was strongly associated with gut microbiota perturbation. We observed a reduced gut microbiota richness in dual-insulted mice with a core microbiota depleted of common gut colonizers, bacteria with probiotic activity, and butyrate producers within the *Clostridium* cluster IV, namely *Butyricicoccus pullicaecorum* and *Papillibacter cinnamivorans*. These two bacterial species colonize the mucus layer closely with the epithelium and, as butyrate producers, may be crucial for achieving epithelial barrier integrity and mediating immunomodulatory effects [37,38]. Microbial communities of 6-OHDA+CFX mice were also depleted in *Bifidobacterium breve* and *Alistipes finegoldii*, main acetate producers, suggesting a lack of cross-feeding interactions with the acetate-depending butyrate producers *B. pullicaecorum* and *P. cinnamivorans*. We also attempted to identify keystone species driving the microbiota dynamics in 6-OHDA+CFX mice. Network analysis built on LEfSe results indicated that the reduction of *A. furcosa, P. cinnamivorans*, and *Defluviitalea saccarophila* drove the loss of the microbiota structure in 6-OHDA+CFX mice, being responsible for the biodiversity decrease. Furthermore, *P. cinnamivorans, A. furcosa, B. breve*, and *A. finegoldii* were negatively correlated in the network analysis with *R. lactaris*, recently renamed *Faecalicatena lactaris*, a member of *Clostridium* cluster XIVa, a mucin-degrading commensal [39] that was strongly enriched in 6-OHDA+CFX mice. Although there is a lack of a well-defined role for *R. lactaris*, some studies associate its enrichment with both chronic gut inflammatory disease and neurodevelopmental disorders [40,41,42], suggesting a pro-inflammatory potential for this bacterial species. Here, we can speculate that the increase of *R. lactaris* in dual-insulted mice can compensate for the decrease of butyrate production bacterial networks by reinforcing mucus glycoprotein degradation and that this activity is detrimental to gut barrier integrity.

One of the main features of gut influence on PD progression in this dual model of PD has been the frailty of the microbiota-butyrate-intestinal barrier axis. BuNa has been shown to exhibit anti-inflammatory and epigenetic effects affecting not only intestinal and systemic homeostasis but also brain function [43]. Notably, several studies demonstrated the beneficial effect of BuNa in various experimental models of PD [8,44,45,46,47,48]. Butyrate reduced neurotoxicity and motor deficit in neurotoxin-induced models of PD [8,46,48].

Here, we also studied the effect of BuNa treatment on 6-OHDA or dual-hit challenged mice. Butyrate showed not only neuroprotective effects but also a beneficial application against several gastrointestinal ailments with an inflammatory background [49,50,51]. Butyrate significantly increased goblet cell numbers and Muc2 expression in the intestines of mice, positively impacting gut permeability and inflammation [52]. Therefore, butyrate represents a multi-target compound able to counteract peripheral and central pathogenic mechanisms. This duality of neuroprotective and enteroprotective effects is likely due to some common pathways involved in the dysregulation of neurodegenerative and gastrointestinal disorders.

Repeated administrations of BuNa improved motor deficits, as shown by the reduction of apomorphine-induced contralateral rotations, in both 6-OHDA and 6-OHDA+CFX mice. The behavioral improvement was associated with the reduction of several pathogenic factors in the striatum, including inflammatory enzyme and TH expression, and apoptosis. Moreover, at the peripheral level, BuNa reduced systemic inflammation and endotoxemia and interestingly counteracted the colonic morphological changes and the related inflammatory process triggered by the 6-OHDA insult or by the dual hit. Interestingly, BuNa treatment operated a mucin-driven change in gut microbial ecology in 6-OHDA+CFX mice. We observed an increase in the number of shared species with the Sham group (including *P. cinnamivorans* and *D. saccharophila*), a reduction of *R. lactaris*, and an increase of the mucin-utilizing specialist *Akkermansia muciniphila*. The reported metataxonomic remodeling of gut microbes could be triggered by butyrate-promoted intestinal mucin biosynthesis, which in turn could act as a prebiotic enabling the outgrowth of both *Akkermansia muciniphila* and butyrate producers. This potential mechanism could be responsible for the amelioration of gut integrity and barrier functions. Overall, we observed that the correct microbial re-colonization was hindered in the two-hit mouse model of PD, suggesting that antibiotic pretreatment enhanced pathophysiological traits and that butyrate treatment helped to establish a new microbial balance with positive effects on PD phenotype.

## 4. Materials and Methods

### 4.1. Chemicals

BuNa, 6-OHDA, and apomorphine were purchased from Sigma-Aldrich (Milan, Italy).

### 4.2. Animals

Ten-week-old male Swiss CD1 mice (20–25 g) were purchased from Harlan (Udine, Italy). They were housed in cages in a room kept at 22 ± 1 °C on a 12/12 h light/dark cycle. The animals were acclimated to their environment for 1 week and had ad libitum access to tap water and standard rodent chow.

### 4.3. Experimental Design and Drug Treatment

The experimental procedure to induce PD was based on the unilateral intrastriatal injection of 6-OHDA (4 µg/2 µL) in mice. Mice were anaesthetized (100 mg/kg ketamine plus 5 mg/kg xylazine intraperitoneally, i.p.) and placed in a stereotaxic apparatus (David Kopf Instruments, Tujunga, CA, USA) with a mouse adaptor and lateral ear bars. The head skin was cut longitudinally, and mice were injected unilaterally in the right striatum with 2 µL of 6-OHDA (2 µg/µL) solution by an injector connected with a polyethylene tubing to a 2 µL Hamilton syringe. 6-OHDA was diluted in 0.9% sterile saline with 0.2% ascorbic acid and filtered inside a laminal flow hood to avoid contamination and protected from light throughout the procedure. Sham mice were injected with the same volume of 0.9% sterile saline and 0.2% ascorbic acid. The following stereotaxic coordinates were used: +0.5 mm anterior to bregma, −2.0 mm lateral to midline, 3.5 mm ventral from the skull surface [53]. Once the 2 µL had been injected, the syringe was kept in place for 5 min before being very slowly retracted from the brain in no less than 5 min.

To induce gut microbiota dysbiosis, mice were treated with ceftriaxone (8 g/kg, per os) once daily, as already reported [30], and five days later, mice were unilaterally injected with 6-OHDA in the right striatum [29]. Starting from the neurotoxin challenge, mice were treated with BuNa (100 mg/kg per os) once daily for 14 days. Therefore, several experimental groups were obtained: (1) Sham control mice receiving intrastriatal injection of the vehicle; (2) 6-OHDA, mice challenged with 6-OHDA; (3) 6-OHDA+BuNa, mice challenged with 6-OHDA and treated with BuNa; (4) 6-OHDA+CFX, mice receiving CFX for 5 days and 6-OHDA intrastriatal injection; (5) 6-OHDA+CFX+BuNa, mice receiving CFX, 6-OHDA intrastriatal injection, and BuNa treatment; and 6) CFX, mice receiving solely the antibiotic for 5 days.

The treatment (BuNa or vehicle) began 3 h after surgery and continued for 14 days. The vehicle was administered to the Sham, 6-OHDA, CFX, and 6-OHDA+CFX groups. Last administrations were performed 2 h before killing. On days 3, 7, and 14 post-6-OHDA injection, behavioral tests (rotarod and apomorphine test) were performed in all groups. At 14 days, feces were collected, mice were sacrificed, and striata and colon were excised and collected to perform later determinations. The experimental protocol is reported in Figure 1A.

### 4.4. Rotarod Test

To evaluate motor activity of mice, we used the rotarod test, which measures balance, coordination, and motor control. The rotarod apparatus (Ugo Basile, Varese, Italy) consists of a suspended rod able to run at constant or accelerating speed. Rotarod maximal rpm was 40.0, and the acceleration time was 200 s. All mice were exposed to a 5 min training period, at accelerating speed, before the neurotoxin injection to familiarize themselves with the apparatus. During the test, mice had to remain on the rod for as long as they could. The length of time that the animal remained on the rod was recorded (a 300 s maximal time was used for the test).

### 4.5. Apomorphine Test

Apomorphine-induced rotation reflects hypersensitivity of the lesioned striatum, and this was assessed by testing over a 45 min test session after challenge with 0.1 mg/kg subcutaneously (s.c.) of apomorphine, dissolved in 0.9% sterile saline with 0.2% ascorbic acid. Animals were allowed to habituate for 5 min after injection before the recording of rotations began.

### 4.6. Western Blot Analysis

Striatum samples were homogenized in ice-cold lysis buffer (20 mM Tris–HCl (pH 7.5), 10 mM NaF, 150 mM NaCl, 1% Nonidet P-40, 1 mM phenylmethylsulfonyl fluoride, 1 mM Na_3_VO_4_, leupeptin and trypsin inhibitor 10 μg/mL; 0.25/50 mg tissue). After 1h, tissue lysates were obtained by centrifugation at 20,000 g for 15 min at 4 °C. Protein concentrations were estimated by the Bio-Rad protein assay (Bio-Rad Laboratories, Milan, Italy) using bovine serum albumin as a standard.

Lysate proteins were dissolved in Laemmli sample buffer, boiled for 5 min, separated by SDS-polyacrylamide gel electrophoresis, and transferred to nitrocellulose membrane (240 mA for 40 min at room temperature). The filter was then blocked with 1 × phosphate buffer saline (PBS) and 5% non-fat dried milk for 40 min at room temperature and probed with anti-cyclooxygenase (COX)-2 (dilution 1:1000; cat. no. 610204, BD Bioscience, from Becton Dickinson, Buccinasco, Italy); anti-inducible nitric oxide synthase (iNOS) antibody (dilution 1:1000; cat. no. 610432, BD Bioscience); anti-nNOS (dilution 1:1000; cat. no. 610308, BD Bioscience); anti-Bcl2 (dilution 1:500, cat. no. sc-492, Santa Cruz Biotechnology, Santa Cruz, CA, USA); anti-Bax (dilution 1:500; cat. no. sc-526, Santa Cruz Biotechnology); or anti-TH (dilution 1:1000; cat. no. AB152, EMD Millipore Corporation) in 1 × PBS, 3% non-fat dried milk and 0.1% Tween 20 at 4 °C overnight. The secondary antibody was incubated for 1 h at room temperature. Subsequently, the blot was extensively washed with PBS, developed using enhanced chemiluminescence detection reagents (Amersham Pharmacia Biotech, Piscataway, NJ, USA) according to the manufacturer’s instructions. The immune complex was visualized by a ChemiDoc Imaging System (Bio-Rad Laboratories). To ascertain that blots were loaded with equal amounts of protein lysates, they were also incubated in the presence of the antibody against the β actin (cat. no. A5441, Sigma-Aldrich, Milan, Italy).

### 4.7. Serum Parameters

Blood was collected and serum was obtained by centrifugation. Serum TNF-α, IL-1β, and IFN-γ were measured using commercially available ELISA kits (Thermo Fisher Scientific, Rockford, IL, USA), following the manufacturer’s instructions. LPS was measured using the Limulus amebocyte lysate (LAL QCL-1000; Lonza Group Ltd., Basel, Switzerland) technique.

### 4.8. Real-Time PCRA

Total RNA was extracted from the colon using Reagent (Bio-Rad Laboratories). Each sample contained 500 ng cDNA in 2X QuantiTect SYBR Green PCR Master Mix, and primers were performed with a Bio-Rad CFX96 Connect Realtime PCR System instrument and software (Bio-Rad Laboratories). The used primers were mouse Cox-2 (Ptgs2), IL-1β (Il1b), and IL-10 (Il10) (Qiagen, Hilden, Germany), and the PCR conditions were previously described [54,55]. The relative amount of each studied mRNA was normalized to GAPDH as a housekeeping gene, and the data were analyzed according to the 2^–ΔΔCT^ method.

### 4.9. Histopathological Analysis

Distal colon tissues were preserved in 10% neutral buffered formalin (code no. 05-01007Q, Bio-Optica, Milan, Italy), dehydrated and embedded in paraffin (code no. 06-7920, Bio-Optica). Tissue sections were stained with hematoxylin and eosin (HE) for morphological studies. Histological specimens were examined and photographed with a light microscope (Nikon Eclipse E600) associated with a microphotography system (Nikon digital camera DMX1200). The histological score was determined according to previous reports [56,57] as follows: % tissue damage (0: no tissue damage, 1: 1–25% damaged tissue, 2: 26–50% damaged tissue, 3: 51–75% damaged tissue, 4: 76–100% damaged tissue), extent of tissue damage (0: no tissue damage, 1: mucosal damage, 2: mucosal and submucosal damage, 3: damage beyond the submucosa), degree of inflammation (0: no inflammation, 1: slight inflammation, 2: moderate inflammation, 3: severe inflammation), extent of crypt damage (0: no crypt damage, 1: basal 1/3 showed damage, 2: basal 2/3 showed damage, 3: only the surface epithelium was intact, 4: the entire crypt and epithelium were lost). The final score was established by summing the scores for each parameter.

### 4.10. Gut Microbiota-Based Studies

Samples for microbiota analysis were collected from a subset of six mice randomly selected from each group and quickly stored at −80 °C. Gut microbiota was studied by performing V3-V4 16S rDNA amplicon sequencing as previously described [58]. Demultiplexed paired-end reads from MiSeq (2 × 300 bp) were processed and analyzed using the Quantitative Insights Into Microbial Ecology (QIIME2, version 2021.4) [59]. Raw FASTQ reads were quality filtered (i.e., filtered, dereplicated, denoised, merged, and assessed for chimeras) to produce amplicon sequence variants (ASV) using the DADA2 pipeline. After phylogenetic tree generation, the Greengenes reference database [60] at a confidence threshold of 99% was used to classify ASVs at different taxonomic levels. In addition, to implement the taxonomic classification at the species level, a representative FASTA sequence of each ASV from DADA2 was aligned to the rRNA/ITs BLAST database [61]. Rarefaction curves were used to estimate the completeness of microbial communities sampling; to avoid sample-size biases in subsequent analyses, a sequence rarefaction procedure was applied using a depth of 2344 sequences/sample. Alpha (Good’s coverage, Shannon’s diversity index, and observed features) and beta diversity metrics (unweighted and weighted UniFrac distances) were employed to study the intra- and inter-group diversity of bacterial communities, respectively. Moreover, group significances between alpha and beta diversity indexes were calculated with QIIME2 plugins using the Kruskal–Wallis test and Analysis of similarities (ANOSIM), respectively. The QIIME2 script “qiime feature-table core-features” was used to identify the core microbiota of each group as the list of genera observed in minimum 50% samples/group. Genera shared between groups were identified in a Venn diagram using Interactive Venn, (http://www.interactivenn.net/ (accessed on 9 September 2021)) [62]. Additionally, the linear discriminant analysis effect size (LEfSe) method [63]; https://huttenhower.sph.harvard.edu/galaxy (accessed on 1 November 2021)) was used to identify genera that differed in relative abundance between groups, i.e., CFX or 6-OHDA insults (LEfSe; *p* < 0.05 by Kruskal–Wallis test, *p* < 0.05 by pairwise Wilcoxon test and logarithmic LDA score of 2.0). Microbial interactions were explored by generating a Spearman co-occurrence network based on the relative abundances of species belonging to key genera. The network was generated using the CoNet plugin [64] for Cytoscape (3.9.0, [65]) by applying the following parameters: nonparametric Spearman correlation coefficients with a minimal cut-off threshold of 0.3 (*p* < 0.05, Bonferroni corrected). At the node level, the following topological features were also considered: betweenness centrality (a measure of centrality based on the number of the shortest paths through each node) and closeness centrality (a measure of centrality based on the possibility for a node to communicate with others depending on a minimum number of shortest paths).

### 4.11. Statistical Analysis

Data are presented as mean ± SEM. All experiments, except for microbiota data processing, were analyzed using analysis of variance (ANOVA) for multiple comparisons followed by Bonferroni’s post hoc test using GraphPad Prism 9 (GraphPad Software, San Diego, CA, USA). Bonferroni’s post hoc test was run only when F was significant. Differences among groups were considered significant at values of *p* < 0.05.

## 5. Conclusions

The designed dual-hit model involves both central and peripheral triggers, exhibiting the synergistic effects of dysbiosis and neuroinflammation in the progression of PD. Here, we proposed a simple modification of a well-standardized PD model in mice by short-term, oral antibiotic pretreatment, to mimic dysfunctional gut-associated mechanisms preceding PD onset. The feasibility and construct validity of this dual-hit model expands its applicability for preclinical studies. The bidirectional communication of the microbiota-gut-brain axis, shown in this model, makes this new paradigm a useful tool for testing or repurposing new compounds, affecting gastrointestinal inflammation and/or microbial dysbiosis, and limiting PD behavioral deficits.

## Figures and Tables

**Figure 1 ijms-23-06367-f001:**
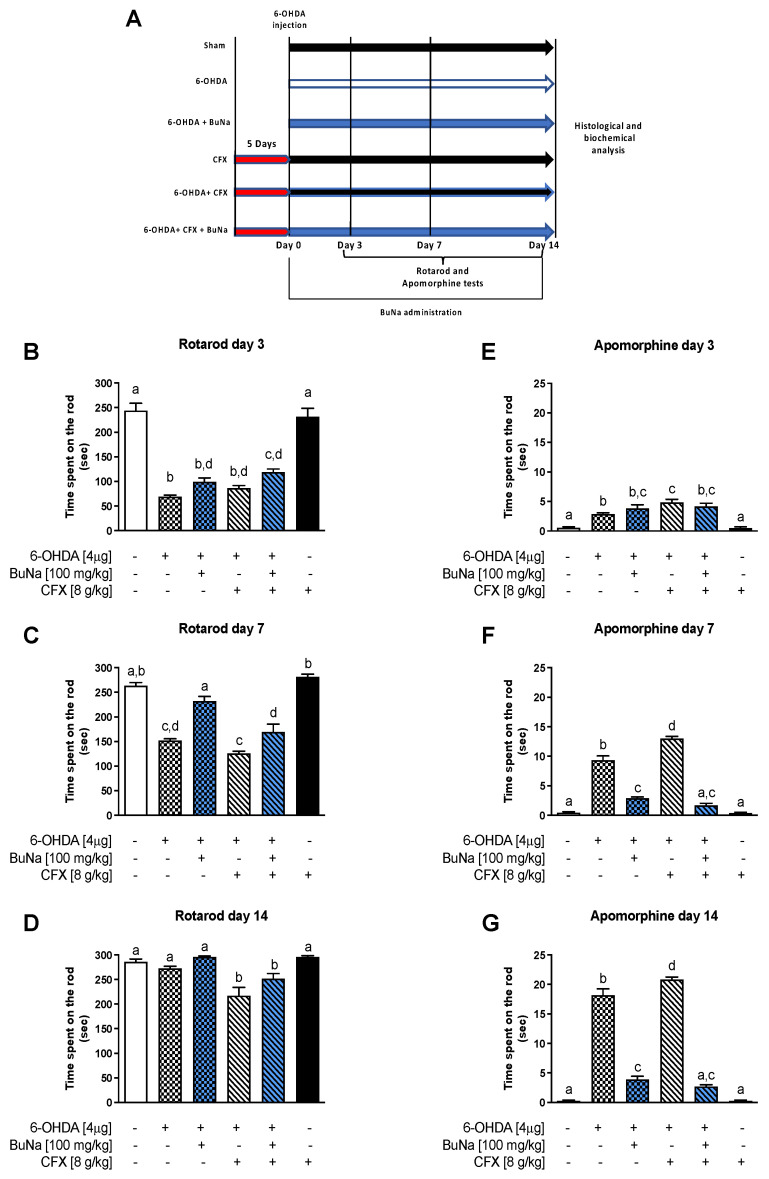
Grouping and schedule (**A**). Effects of BuNa (100 mg/kg) on motor coordination in 6-OHDA-lesioned mice pretreated or not with CFX (**B**–**D**). CD1 mice were treated with BuNa daily, and time spent on the rod (sec) was assessed and recorded at 3 (**A**), 7 (**C**), and 14 (**D**) days. Values are expressed as mean ± SEM of retention time on the rotating bar. Evaluation of apomorphine-induced rotational behavior (**E**–**G**). The number of net ipsilateral and contralateral rotations was counted per minute in all animals from all groups after 3 (**E**), 7 (**F**), and 14 (**G**) days. Values are expressed as mean ± SEM (*n* = 8–12) of all turns collected during 40 min. Labeled means without a common letter differ, *p* < 0.05.

**Figure 2 ijms-23-06367-f002:**
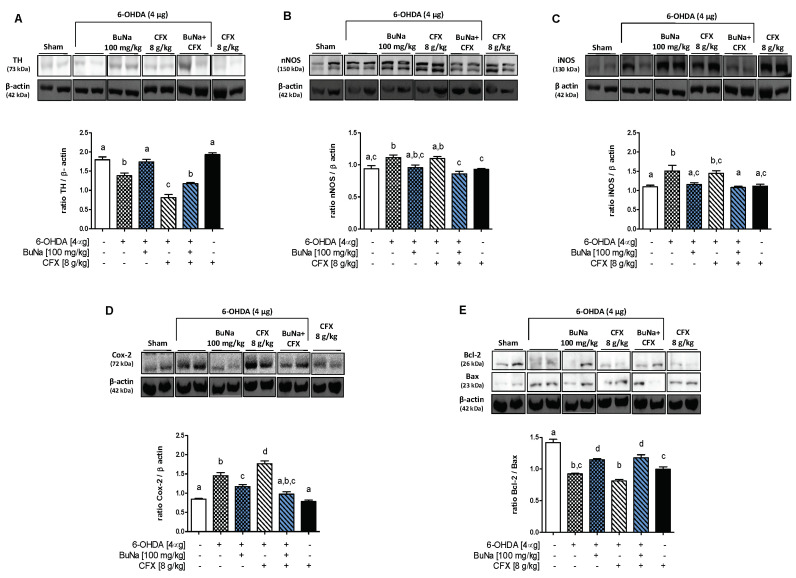
Neuroprotective effects of BuNa (100 mg/kg) measured using tyrosine hydroxylase, inflammatory and apoptotic markers in the striatum in mice: (**A**) TH (**B**) nNOS, (**C**) iNOS (**D**) COX-2, (**E**) Bcl-2 and Bax expression reported as the ratio of optical densities of their bands. Representative immunoblots of all tissues analyzed were shown. Densitometric analysis of protein bands is reported: the levels are expressed as the density ratio of target to control protein bands (β-actin). Values are expressed as mean ± SEM (*n* = 4–6). Labeled means without a common letter differ, *p* < 0.05.

**Figure 3 ijms-23-06367-f003:**
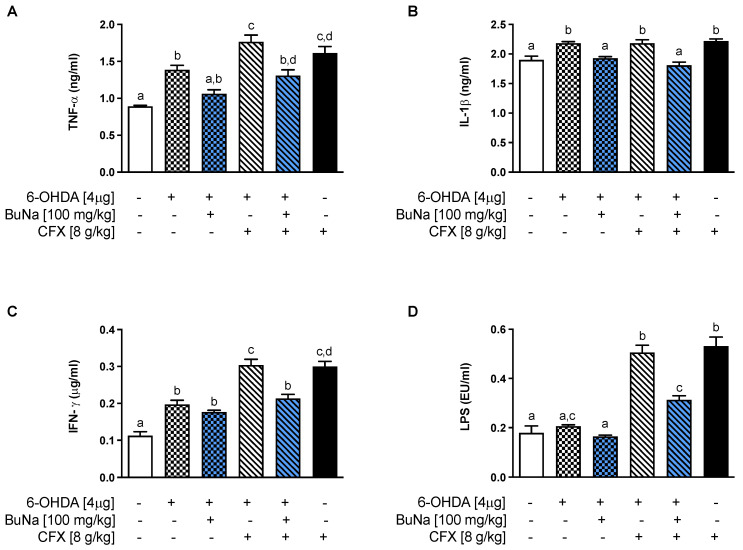
BuNa-modulated systemic parameters of the immune system and inflammation altered by 6-OHDA and CFX. BuNa-limited (**A**) TNF-α, (**B**) IL-1β, (**C**) IFN-γ, and (**D**) LPS increased in 6-OHDA and/or CFX+6-OHDA mice. All data are shown as mean ± S.E.M (*n* = 6 per group). Labeled means without a common letter differ, *p* < 0.05.

**Figure 4 ijms-23-06367-f004:**
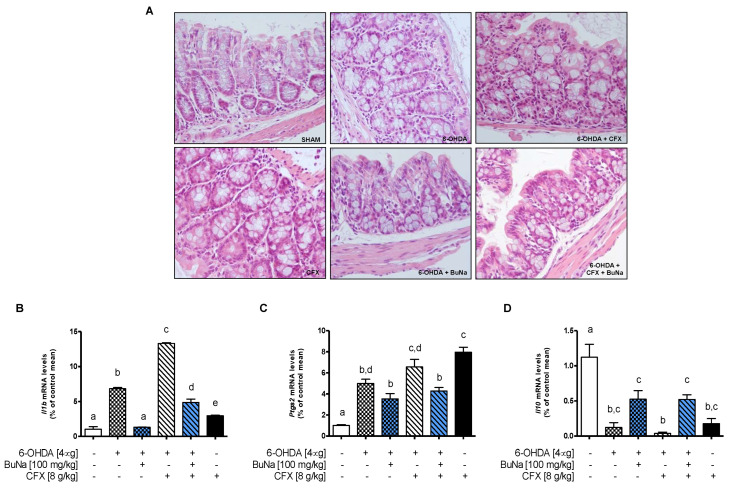
Histological analysis of distal colon sections of mice. (**A**) Haematoxylin & eosin staining (magnification ×400). (**B**–**D**) Transcriptional levels of *Il1b*, *Ptgs2*, and *Il10* were evaluated in colonic tissues. All data are shown as mean ± S.E.M (*n* = 4–6). Labeled means without a common letter differ, *p* < 0.05.

**Figure 5 ijms-23-06367-f005:**
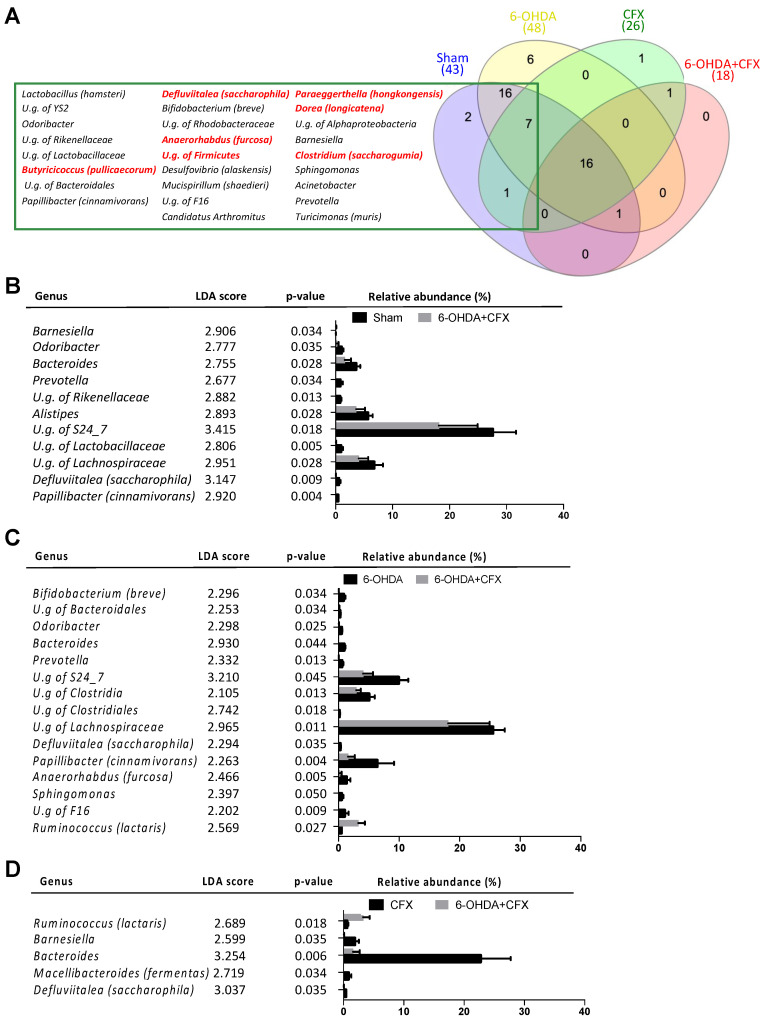
Antibiotic-induced dysbiosis in 6-OHDA-lesioned mice. (**A**) Venn diagram illustrating bacterial genera overlaps among core microbiomes with an attached list of the 26 genera missing in 6-OHDA+CFX to Sham group; in red are the 7 bacterial genera that specifically failed to recover in 6-OHDA+CFX. (**B**–**D**) Gut microbiota differences at genus taxonomic level based on linear discriminant analysis (LDA) combined with effect size (LEfSe) algorithm (*p* > 0.05 for both Kruskal–Wallis and pairwise Wilcoxon tests and a cutoff value of LDA score above 2.0). In each panel, LDA scores, *p*-values, and relative abundance of key genera discriminating bacterial communities are reported, when possible, and the species taxonomic classification/genus is also reported.

**Figure 6 ijms-23-06367-f006:**
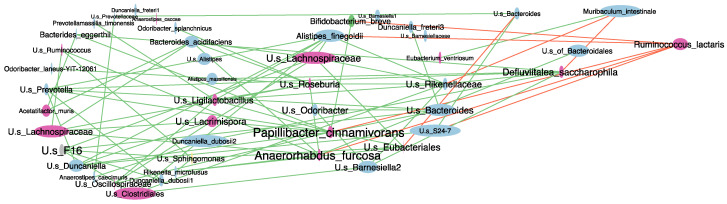
Co-occurrence network of species belonging to key genera discriminating Sham, 6-OHDA, CFX, and 6-OHDA+CFX mice. In the networks: the nodes represent the species, and the edges show nonparametric Spearman correlations with a correlation coefficient >0.5 or ≤0.5 that is statistically significant (*p* < 0.05 after Bonferroni correction), within green copresence interaction type, while in red mutual exclusion interaction type; node and label sizes are proportional to the relative abundance and its degree (the number of edges and nodes connected to each node), respectively. The colors of nodes represent their classification at the phylum level (green: Actinobacteria; blue: Bacteroidetes; pink: Firmicutes; yellow: Proteobacteria; grey: TM7).

**Figure 7 ijms-23-06367-f007:**
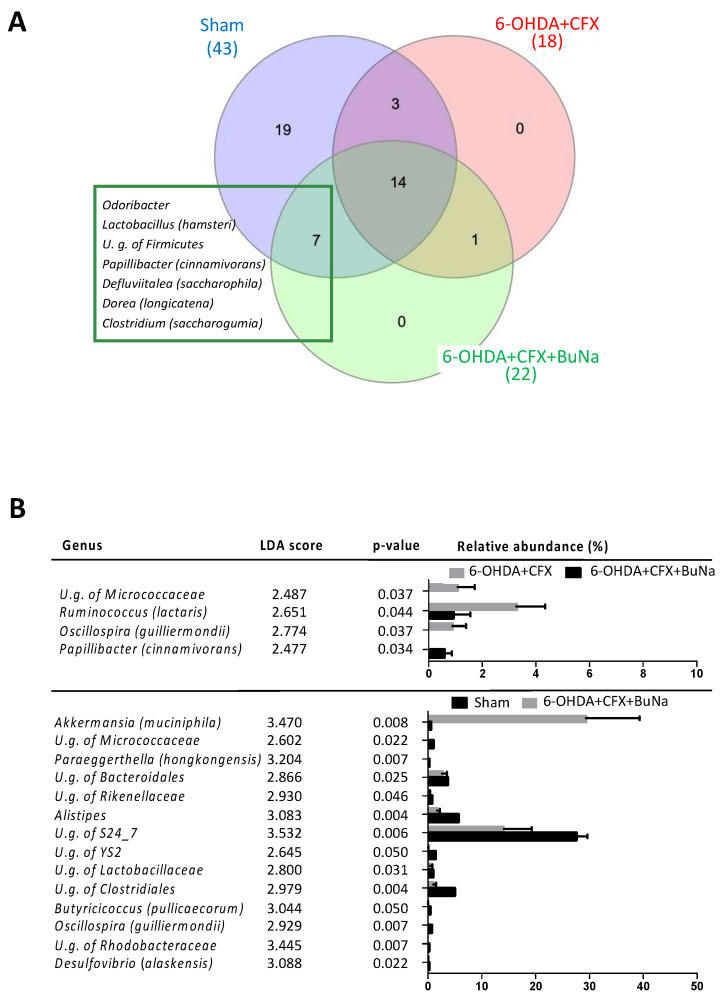
BuNa-restructured gut microbiota in 6-OHDA+CFX mice. (**A**) Venn diagram illustrating bacterial genera overlaps among core microbiomes with an attached list of the 7 genera shared by Sham and 6-OHDA+CFX+BuNa mice and missing in 6-OHDA+CFX mice. (**B**) Gut microbiota differences at genus taxonomic level based on linear discriminant analysis (LDA) combined with effect size (LEfSe) algorithm (*p* > 0.05 for both Kruskal–Wallis and pairwise Wilcoxon tests and a cutoff value of LDA score above 2.0). LDA scores, *p*-values, and relative abundance of key genera discriminating bacterial communities are reported in each panel.

## Data Availability

Not applicable.

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
