# Peer review of "Dual-Hit Model of Parkinson’s Disease: Impact of Dysbiosis on 6-Hydroxydopamine-Insulted Mice—Neuroprotective and Anti-Inflammatory Effects of Butyrate"

_ijms, 2022, doi:10.3390/ijms23126367_

Round 1

Reviewer 1 Report

The manuscript entitled (Dual-hit model of Parkinson’s disease: impact of dysbiosis on 6-hydroxydopamine-insulted mice. Neuroprotective and anti- inflammatory effects of butyrate) by Avagliano et al. presented a two-hit model of PD through ceftriaxone (CFX)-induced dysbiosis and gut inflammation before the 6-hydroxydopamine (6-OHDA) intrastriatal injection to mimic dysfunctional gut-associated mechanisms preceding PD onset. Overall, the manuscript is well-prepared and the discussion is appropriate. However, some issues should be considered prior to publication:

  1. The introduction section does not present enough background about the relationship between gut dysbiosis and inflammation in PD progression.
  2. A bio-figure demonstrating the theoretical background of the proposed research will be a good addition to this paper.
  3.  In Figures 5, B, C, and D, the font is not appropriate for readers 
  4. In the introduction, I recommend adding a reference related to the recent Emerging Therapeutic Strategies for Parkinson’s Disease and the Future Prospects (doi.org/10.3390/biomedicines10020371)
  5. Regarding the conclusion, I recommend authors mention a small paragraph about how their proposed model could be an applicable technique in PD preclinical studies.
  6. Finally, the manuscript should be revised by a native English speaker and the abstract should be presented as a whole paragraph. 

Author Response

  1. The introduction section does not present enough background about the relationship between gut dysbiosis and inflammation in PD progression.

We thank the referee for the convenient suggestion. Some sentences were added to Introduction section to elucidate this issue (Lines 77-88).

  1. A bio-figure demonstrating the theoretical background of the proposed research will be a good addition to this paper.

Graphical abstract was added in the manuscript.

  1. In Figures 5, B, C, and D, the font is not appropriate for readers

We thank the reviewer for the comment. We agree that the figure as it stood was not clear. The font was upgraded.

  1. In the introduction, I recommend adding a reference related to the recent Emerging Therapeutic Strategies for Parkinson’s Disease and the Future Prospects (doi.org/10.3390/biomedicines10020371)

The review article was quoted in Introduction section (lines 71-74).

  1. Regarding the conclusion, I recommend authors mention a small paragraph about how their proposed model could be an applicable technique in PD preclinical studies.

The strength and applicability of this model were described in the Conclusions (lines 607-610).

  1. Finally, the manuscript should be revised by a native English speaker and the abstract should be presented as a whole paragraph.

Thanks to the suggestion from reviewer, we extensively revised the manuscript. English grammar was checked, and the abstract was re-arranged as requested.

Reviewer 2 Report

Summary The manuscript by Avagliano et al is interesting and important for the field. Accurate pathology need to be replicated in animal models for pre-clinical studies to be accurate and relevant. The manuscript uses 6-OHDA coupled with ceftriaxone to induce a dual hit PD model. The authors study Tyrosine hydroxylase levels, inflammatory markers, apoptotic markers, perform behavioral assay and characterize gut microbiota. The manuscript is very well written.    Comments
  • Line 78-79:  please specify "impairment of PD animal models", as 12 and 13 references are based on PD animal models.
  • The know molecular mechanism of action of Sodium butyrate should be addressed in the introduction section.
  • The experimental protocol (figure 8 ) should be placed as a first figure to aid in better understanding of drug dosage and regimen.
  • Line 97: mices coordination or motor coordination?
  • Generally , all the figures need work. The text is hard to read and the figures should be bigger and legible. 
  • Statistics should be mentioned in the figure legend. 
  • A major shortcoming is the choice of technique in the form of western blotting based quantification of Tyrosine hydroxylase levels in striatum, which restricts information on dopaminergic neuronal loss. As it is evident that unique dopaminergic neuronal populations are selectively effected in human as well as PD models. Cell lysates from striatum fail to account for the neuronal diversity and thereby diluting the real effect. The better option and field accepted technique would be TH or DAT based immunohistochemistry in SNPC of the mouse striatum sections. 
  • The western blot images are too small and tubulin blots are especially are not very clear. The authors need to improve the WB figures. 
  • Results section 2.3- Its is not mentioned what technique was used to measure inflammatory cytokines
  • The scoring methodology to access colon injury can be described in result section for better reading experience. 

Author Response

Line 78-79:  please specify "impairment of PD animal models", as 12 and 13 references are based on PD animal models.

As requested, we have specified “in animal models” (line 88)

 The know molecular mechanism of action of Sodium butyrate should be addressed in the introduction section.

A sentence was added on the multiple molecular targets of sodium butyrate, as requested (lines 98-104).

The experimental protocol (figure 8) should be placed as a first figure to aid in better understanding of drug dosage and regimen.

Figure 8 was placed as first figure, as suggested (See figure 1A and line 478).

Line 97: mices coordination or motor coordination?

The mistake was corrected (line 115).

Generally, all the figures need work. The text is hard to read and the figures should be bigger and legible.

All the figures were revised, according to journal guidelines. Also the text was improved.

Statistics should be mentioned in the figure legend.

The sentence “Labeled means without a common letter differ, P < 0.05” was added in all figure Legends.

A major shortcoming is the choice of technique in the form of western blotting based quantification of Tyrosine hydroxylase levels in striatum, which restricts information on dopaminergic neuronal loss. As it is evident that unique dopaminergic neuronal populations are selectively effected in human as well as PD models. Cell lysates from striatum fail to account for the neuronal diversity and thereby diluting the real effect. The better option and field accepted technique would be TH or DAT based immunohistochemistry in SNPC of the mouse striatum sections.

Based on the small number and low amount of striatal tissues in order to meet the principle of the 3Rs, we opted for Western blot analysis to evaluate several features, i.e. inflammation (COX-2, iNOS, nNOS), apoptosis (Bax, Bcl-2), and dopaminergic loss (TH). We have considered that WB was a versatile technique to evaluate at once all those parameters through protein expression. Indeed, regarding the selectivity of dopaminergic neuronal loss, apart from TH expression, we thought that this issue was also corroborated by the functional analysis we showed through apomorphine test, that, as written in the manuscript, reflects dopamine receptor hypersensitivity because of dopamine depletion and dopaminergic neuronal loss.

The western blot images are too small and tubulin blots are especially not very clear. The authors need to improve the WB figures.

Western blot images were improved to the best, accordingly with journal guidelines.

Results section 2.3- Its is not mentioned what technique was used to measure inflammatory cytokines

Indeed, the technique for cytokine quantification was reported in Materials and Methods section (lines 524-529).

The scoring methodology to access colon injury can be described in result section for better reading experience.

Thank you for the suggestion. We have now reported in Results section (2.4 paragraph) the parameters considered to score the overall colon injury in the histological analysis (lines 191-196).

Round 2

Reviewer 1 Report

All comments have been handled in a positive manner